# Protein Unfolding—Thermodynamic Perspectives and Unfolding Models

**DOI:** 10.3390/ijms24065457

**Published:** 2023-03-13

**Authors:** Joachim Seelig, Anna Seelig

**Affiliations:** Biozentrum, University of Basel, Spitalstrasse 41, CH-4056 Basel, Switzerland

**Keywords:** protein unfolding, differential scanning calorimetry, Zimm–Bragg theory, cold denaturation, free energy

## Abstract

We review the key steps leading to an improved analysis of thermal protein unfolding. Thermal unfolding is a dynamic cooperative process with many short-lived intermediates. Protein unfolding has been measured by various spectroscopic techniques that reveal structural changes, and by differential scanning calorimetry (DSC) that provides the heat capacity change C_p_(T). The corresponding temperature profiles of enthalpy ΔH(T), entropy ΔS(T), and free energy ΔG(T) have thus far been evaluated using a chemical equilibrium two-state model. Taking a different approach, we demonstrated that the temperature profiles of enthalpy ΔH(T), entropy ΔS(T), and free energy ΔG(T) can be obtained directly by a numerical integration of the heat capacity profile C_p_(T). DSC thus offers the unique possibility to assess these parameters without resorting to a model. These experimental parameters now allow us to examine the predictions of different unfolding models. The standard two-state model fits the experimental heat capacity peak quite well. However, neither the enthalpy nor entropy profiles (predicted to be almost linear) are congruent with the measured sigmoidal temperature profiles, nor is the parabolic free energy profile congruent with the experimentally observed trapezoidal temperature profile. We introduce three new models, an empirical two-state model, a statistical–mechanical two-state model and a cooperative statistical-mechanical multistate model. The empirical model partially corrects for the deficits of the standard model. However, only the two statistical–mechanical models are thermodynamically consistent. The two-state models yield good fits for the enthalpy, entropy and free energy of unfolding of small proteins. The cooperative statistical–mechanical multistate model yields perfect fits, even for the unfolding of large proteins such as antibodies.

## 1. Introduction

Many proteins are only marginally stable at room temperature and can be denatured by heating or cooling. The analysis of protein stability is thus an important problem in developing biological therapeutics. The protein folding–unfolding reaction is a dynamic equilibrium between many different short-lived intermediates. This was recognized as early as 1959 by B. Zimm and J.K. Bragg, who published a seminal theory on the helix-coil phase transition of polypeptides [1]. The complex statistical–mechanical theory was explained in simple terms in a textbook by N. Davidson in 1962 [2]. The Zimm–Bragg theory is not limited to structural changes but was also essential in determining the kinetics of the helix-coil transition in 1968 [3]. The Zimm–Bragg theory follows from the linear Ising model of magnetism [4], and its physics is somewhat demanding. It is probably for this reason that the multistate cooperative theory which takes into account that molecular elements (e.g., amino acid residues) act dependently on each other was largely ignored by experimentalists in protein unfolding. Instead, a much simpler two-state model for protein unfolding became the popular and almost exclusive alternative. A two-state model considers only two types of protein conformations in solution, the native protein (N) and the fully unfolded protein (U). No molecular interactions are specified in a two-state model, which therefore must be classified as non-cooperative.

The chemical equilibrium two-state model was originally proposed by Brandts in 1964 [5,6] He measured the thermal unfolding of chymotrypsinogen with spectroscopic techniques and defined an equilibrium constant as the ratio of unfolded-to-native protein for each temperature. The van’t Hoff plot of the temperature-dependent equilibrium constant provided the unfolding enthalpy ΔH_0_. In this early model, the unfolding enthalpy was temperature-independent. A direct measurement of the unfolding enthalpy was made possible by differential scanning calorimetry (DSC) (for details of differential scanning calorimetry (DSC), see references [7,8,9]. DSC measures the heat capacity C_p_(T) that is the temperature derivative of the enthalpy change ΔH(T) at constant pressure p. The relevant DSC literature is focused on the simulation of the heat capacity peak associated with protein unfolding. The temperature-induced thermodynamic unfolding parameters ΔH(T), ΔS(T) and ΔG(T) have thus far been derived in an exclusively model-guided manner (see [10]). A historical perspective of the chemical equilibrium two-state model, used for this purpose, can be found in [11,12].

We have recently demonstrated that DSC can achieve more. It directly provides the thermodynamic unfolding parameters ΔH(T), ΔS(T) and ΔG(T) from the heat capacity C_p_(T) measurement, independent of any model [13,14]. Knowledge of these experimental parameters now allows for the examination of different models.

In this review, we show how the thermodynamic functions enthalpy ΔH(T), entropy ΔS(T) and free energy ΔG(T) can be calculated by numerical integration of C_p_(T) without the application of a protein-unfolding model. As an example, we use the heat denaturation of lysozyme (Section 2.1). In Section 2.2, we introduce the currently prevailing (or standard) chemical equilibrium two-state model, and three new models [15], the Θ_U_(T)-weighted chemical equilibrium two-state model, the statistical–mechanical two-state model and the statistical–mechanical multistate model [13]. Whereas the first two are empirical models, the latter two are based on rigorous thermodynamic partition functions. These models allow the simulation of C_p_(T) in terms of the enthalpy ΔH(T), entropy ΔS(T) and free energy ΔG(T). In Section 3, we emphasize again the experimental (model-independent) analysis of the heat capacity C_p_(T) in terms of the enthalpy ΔH(T), entropy ΔS(T) and free energy ΔG(T) upon heat denaturation of lysozyme and heat and cold denaturation of β-lactoglobulin and examine whether these data can be fitted with the different models. The standard chemical equilibrium two-state model shows discrepancies with respect to entropy and free energy and casts doubt on the physical reality of the postulated positive free energy of the native protein. The Θ_U_(T)-weighted chemical equilibrium two-state model corrects most of the insufficiencies of the standard model. The two new statistical–mechanical models are based on rigorous thermodynamic partition functions and provide perfect simulations of all measured thermodynamic properties.

## 2. Methods

### 2.1. Differential Scanning Calorimetry (DSC). Model-Independent Thermodynamic Analysis of Protein Unfolding Experiments

Differential scanning calorimetry (DSC) is the method of choice to study the thermodynamic properties of protein unfolding. DSC measures the heat capacity C_p_(T) as a function of temperature. Here, we show that the fundamental thermodynamic properties of protein unfolding can be derived directly by numerical integration of the heat capacity [13,14], as follows:(1)$$\mathrm{enthalpy}\;\Delta H_{\mathrm{DSC}}(T_{i})=\sum_{1}^{i}\left[\frac{C_{p}(T_{i})+C_{p}(T_{i+1})}{2}\right]\left[T_{i+1}-T_{i}\right]$$
(2)$$\mathrm{entropy}\;\Delta S_{\mathrm{DSC}}(T_{i})=\sum_{1}^{i}\left[\frac{C_{p}(T_{i+1})+C_{p}(T_{i})}{2T_{i}}\right]\left[T_{i+1}-T_{i}\right]$$
(3)$$\mathrm{free}\;\mathrm{energy}\;\Delta G_{\mathrm{DSC}}\left(T_{i}\right)=\Delta H\left(T_{i}\right)-T_{i}\Delta S\left(T_{i}\right)$$

Note that all thermodynamic properties can be evaluated without resorting to a particular unfolding model.

Native proteins have a substantial heat capacity [16]. A typical DSC-thermogram is shown in Figure 1A. The heat capacity C_p_(T) starts out almost horizontally, reflecting the basal value of the native protein. Unfolding then gives rise to a sharp heat capacity peak, which is followed again by a region of rather constant heat capacity of the unfolded protein. The heat capacity of the unfolded protein is distinctly higher than that of the native protein ($\Delta C_{p}^{0}$ > 0). $\Delta C_{p}^{0}$ scales with the size of the protein [17,18]. The increase $\Delta C_{p}^{0}$, is caused essentially by the binding of additional water molecules [18].

Unfolding models generally assume baseline corrected thermograms with zero heat capacity for the native protein. Of note, equations, 1–3 are not limited to baseline corrected thermograms. Evaluations which include the substantial heat capacity of the native protein are given in a recent publication on cooperative protein unfolding [13].

The choice of the DSC-baseline is important and is handled quite differently in the literature [8]. The subtraction of a sigmoidal baseline has become quite common such that the heat capacity difference $\Delta C_{p}^{0}$ between the native and denatured protein is lost [19]. If the increase in the basic heat capacity $\Delta C_{p}^{0}$ of the unfolded protein is ignored, the further analysis is limited to the conformational change only. This approach has been criticized as follows: “It is clear that in considering the energetic characteristics of protein unfolding one has to take into account all energy which is accumulated upon heating and not only the very substantial heat effect associated with gross conformational transitions, that is, all the excess heat effects must be integrated” [7].

The analysis of DSC experiments as performed in this review is shown in Figure 1 for the thermal unfolding of lysozyme [20,21]. Lysozyme is a 129-residue protein composed of ~25% α-helix, ~40% β-structure and ~35% random coil in solution at room temperature [20]. Upon unfolding, the α-helix is almost completely lost and the random coil content increases to ~60%. Thermal unfolding occurs in the range of 43 °C < T < 73 °C and is completely reversible. Lysozyme is the classical example to demonstrate two-state unfolding [10,22,23].

Figure 1A displays the heat capacity C_p_(T) [20,21]. The midpoint of unfolding is at T_m_ = 62 °C. Panels 1B–1D show the summation of C_p_(T_i_) according to Equations (1)–(3). Due to baseline correction, the basal heat capacity of the native lysozyme is removed and the figure shows the heat capacity change of the unfolding transition proper. For the native protein, the heat capacity change upon unfolding is zero (C_p_(T) = 0 cal/molK). According to basic thermodynamics it follows that the change in all thermodynamic properties must also be zero for the native protein.

As shown in Figure 1A, the heat capacity C_p_(T_i_) is a non–linear function of temperature. Consequently, enthalpy Δ$H(T)=\int C_{p}(T)dT$, entropy Δ$S(T)=\int \frac{C_{p}(T)}{T}dT$ and Gibbs free energy ΔG(T)=H(T)−TΔS(T) are also non–linear with temperature T. Indeed, enthalpy and entropy display sigmoidal temperature profiles (Figure 1B,C). The free energy change of lysozyme is zero for the native protein, is slightly negative up to the midpoint temperature T_m_, and decreases rapidly beyond T_m_ (Figure 1D).

The typical heat capacity profiles of protein unfolding (Figure 1A) are well known. However, the corresponding temperature profiles of enthalpy, entropy or free energy, have to our knowledge not been reported in the relevant literature, even though these thermodynamic functions are essential in a model-guided analysis.

### 2.2. Models for Protein Unfolding

#### 2.2.1. Chemical Equilibrium Two-State Models

Protein unfolding is a cooperative process. Nevertheless, in spite of many short-lived intermediates, protein unfolding is almost exclusively described by a chemical equilibrium between a single native protein conformation (N) and a single denatured molecule (U). The temperature-dependent equilibrium constant is defined as
(4)$$K_{\mathrm{NU}}(T)=\frac{\left[U\right]}{\left[N\right]}$$

[U] and [N] denote the concentrations of unfolded and native protein, respectively. The temperature dependence of equilibrium constant K_NU_(T) is handled differently in different models.

(a)Van’t Hoff Enthalpy Model:

The early version of the two-state model is based on van’t Hoff’s law [5,9,17]. The temperature dependence of the chemical equilibrium constant is given by
(5)$$\frac{\partial \ln K_{\mathrm{NU}}(T)}{\partial T}=\frac{\Delta H_{0}}{RT^{2}}$$

ΔH_0_ is the conformational enthalpy, R is the molar gas constant, and T the absolute temperature. Integration of Equation (5) yields $\ln K_{\mathrm{NU}}(T)=\left(-\Delta H_{0}/RT\right)+C$
(6)$$K_{\mathrm{NU}}(T)=e^{\frac{-\Delta H_{0}}{R}\left(\frac{1}{T}-\frac{1}{T_{m}}\right)}$$

The integration constant C was chosen such that the equilibrium constant is unity at the midpoint temperature of unfolding T_m_, that is, K_NU_(T_m_) = 1. At T_m_, native and unfolded protein occur at equal concentrations.

In the van’t Hoff model the enthalpy is temperature-independent and the unfolded protein has the same basic heat capacity as the native protein [9,17,24]. To account for the experimentally observed increase in the heat capacity of the unfolded protein, the van’t Hoff model was replaced by a more general model with a temperature-dependent enthalpy.

(b)Free Energy Chemical Equilibrium Two-State Model (“Standard Model”):

This model is based on the free energy which is temperature dependent [25,26]. The temperature dependence of the N ⇄ U equilibrium is calculated with the free energy ΔG_NU_(T). We follow the common nomenclature [7,26,27]
(7)$$K_{\mathrm{NU}}(T)=\frac{\left[U\right]}{\left[N\right]}=e^{\frac{-\Delta G_{\mathrm{NU}}(T)}{RT}}$$

The free energy ΔG_NU_(T) is composed of the enthalpy ΔH_NU_(T) and the entropy ΔS_NU_(T).
(8)$$\Delta H_{\mathrm{NU}}(T)=\Delta H_{0}+\Delta C_{p}^{0}(T-T_{m})$$

ΔH_0_ is the conformational enthalpy proper and $\Delta C_{p}^{0}$ is the increase in heat capacity between the native and the unfolded protein. The entropy ΔS_NU_(T) is defined as
(9)$$\Delta S_{\mathrm{NU}}(T)=\Delta S_{0}+\Delta C_{p}^{0}\ln\frac{T}{T_{m}}=\frac{\Delta H_{0}}{T_{m}}+\Delta C_{p}^{0}\ln\frac{T}{T_{m}}$$

The conformational entropy $\Delta S_{0}$ is evaluated by assuming a first-order phase transition (e.g., melting of ice). In such a phase transition the total heat ΔH_0_ is absorbed at a constant temperature T_m_ and the entropy change is ΔS_0_ = ΔH_0_/T_m_. With this assumption the free energy follows as
(10)$$\Delta G_{\mathrm{NU}}(T)=-RT\ln K_{\mathrm{NU}}(T)=\Delta H_{0}^{}(1-\frac{T}{T_{m}})+\Delta C_{p}^{0}(T-T_{m})-T\Delta C_{p}^{0}\ln(\frac{T}{T_{m}})$$

It should be noted however, that in protein unfolding ΔH_0_ is absorbed not at a constant temperature but over a temperature range of 20–50 °C.

The extent of unfolding Θ_U_(T) is
(11)$$\Theta_{U}^{}(T)=\frac{K_{\mathrm{NU}}(T)}{1+K_{\mathrm{NU}}(T)}=\frac{e^{\frac{-\Delta G_{NU}(T)}{RT}}}{1+e^{\frac{-\Delta G_{{}_{NU}}(T)}{RT}}}$$

The thermodynamic definition of the heat capacity is $C_{p}(T)={\left(\partial H(T)/\partial T\right)}_{p}$. The heat capacity of the chemical equilibrium two-state model is
(12)$$C_{p}(T)=\frac{\partial \left(\Theta_{U}(T)\Delta H_{NU}(T)\right)}{\partial T}=\Delta H_{\mathrm{NU}}(T)\frac{\partial \Theta_{U}(T)}{\partial T}+\Theta_{U}(T)\Delta C_{p}^{0}$$

It should be noted that the unfolding enthalpy ΔH_NU_(T) is multiplied with the extent of unfolding Θ_U_(T) to account for the non–linear temperature profile of the heat capacity C_p_(T). Equation (12) is identical to Equation (14) in reference [26] and is the hallmark of the standard chemical equilibrium two-state model.

The thermodynamic predictions of this model are shown in Figure 2. The enthalpy is a linear function of temperature, the entropy is almost linear, and the free energy ΔG(T) has the approximate shape of an inverted parabola. The native protein has a free energy maximum of 7.51 kcal/mol at 290 K = 17 °C. However, as the heat capacity is zero at the same temperature this reveals a thermodynamic inconsistency. As shown experimentally in Figure 1, a zero heat capacity change leads to zero values for all thermodynamic functions. The temperatures for heat and cold unfolding are T_m_ = 63 °C and T_cold_ = −24 °C, respectively. At these temperatures, folded and unfolded protein are present at equal concentrations. Cold denaturation may not be feasible experimentally for many proteins, but T_cold_ can be estimated as
(13)$$T_{\mathrm{cold}}\approx T_{m}(2e^{\frac{-\Delta H_{0}}{T_{m}\Delta C_{p}^{0}}}-1)$$

The temperature difference between heat and cold denaturation is
(14)$$\Delta T\approx 2T_{m}(1-e^{\frac{-\Delta H_{0}}{T_{m}\Delta C_{p}^{0}}})$$

ΔT depends essentially on the ratio $\Delta H_{0}/\Delta C_{p}^{0}$. The two parameters have opposite effects. ΔH_0_ increases ΔT, $\Delta C_{p}^{0}$ decreases it.

(c)Θ_U_(T)-Weighted Chemical Equilibrium Two-State Model [15]:

The heat capacity of lysozyme in Figure 1A shows a non–linear temperature profile. This was taken into account in the chemical equilibrium model (Equation (12)) by differentiating the product $\Delta H_{\mathrm{NU}}(T)\Theta_{U}(T)$, not just ΔH_NU_(T) (which would result in a constant heat capacity, Equation (8)). As C_p_(T) is non–linear the other thermodynamic functions must also be non–linear. We therefore applied the empirical approach of Equation (12) not only to enthalpy, but for consistence also to entropy and free energy and defined a new set of Θ_U_(T)-weighted thermodynamic functions.
(15)$$\Delta H_{\Theta }(T)=\Theta_{U}(T)\Delta H_{\mathrm{NU}}(T)$$
(16)$$\Delta S_{\Theta }(T)=\Delta S_{\mathrm{NU}}(T)\Theta_{U}(T)$$
(17)$$\Delta G_{\Theta }(T)=\Delta G_{\mathrm{NU}}(T)\Theta_{U}(T)$$

The Equation (12) for the heat capacity is not repeated here. The resulting temperature profiles are shown in Figure 3.

The weighting factor Θ_U_(T) generates sigmoidal temperature profiles for enthalpy and entropy and a trapezoidal profile for the free energy. The free energy change $\Delta G_{\Theta }(T)$ of the native protein is zero, which is now consistent with the zero heat capacity. Of note, the Θ_U_(T) is an empirical weighting factor, not based on solid thermodynamic reasoning. A closer inspection of Equation (17) reveals residual small positive free energies in the vicinity of the midpoint temperatures T_m_ and T_cold_ (Figure 3D, for details see below, enlargement in view of the free energy). These small positive free energies are not observed experimentally.

#### 2.2.2. Statistical-Mechanical Models

(a)Partition Function Z(T) and Thermodynamic Properties:

Statistical-mechanics provides a rigorous thermodynamic approach to protein unfolding. The heat capacity C_p_(T) is intimately related to the partition function Z(T) (for definition see Section 2.2.2). Knowledge of the partition function Z(T) then leads to the following thermodynamic relations [28,29]
(18)Helmholtz free energy: F(T)=−RTlnZ(T)
(19)$$\mathrm{Inner}\;\mathrm{energy}:\;E(T)=RT^{2}\frac{\partial \ln Z(T)}{\partial T}$$
(20)$$\mathrm{Entropy}:\;S_{v}(T)=\frac{E(T)-F(T)}{T}$$
(21)$$\mathrm{Heat}\;\mathrm{capacity}:\;C_{V}(T)={\left(\frac{\partial E(T)}{\partial T}\right)}_{V}=\frac{<E{\left(T\right)}^{2}>-<E(T){>}^{2}}{RT^{2}}$$

DSC experiments are made at constant pressure. As the volume changes in protein unfolding are very small (<5%), the following relations are applicable without loss of accuracy: heat capacity C_p_(T) ≅ C_v_(T), enthalpy H(T) ≅ inner energy E(T), entropy S_p_(T) ≅ S_v_(T), Gibbs free energy G(T) ≅ Helmholz free energy H(T) [13].

(b)Statistical-Mechanical 2-State Unfolding Model [15]:

The problem is to find the partition function of a one-component two-state system. Based on the statistics of the linear Ising model as described in reference [2], the following continuous canonical partition function can be defined
(22)$$Z(T)=\left(1+e^{\frac{-\left[\Delta E_{0}+C_{v}(T-T_{m})\right]}{R}\left[\frac{1}{T}-\frac{1}{T_{m}}\right]}\right)$$

ΔE_0_ is the conformational energy of the unfolded protein. It is temperature-dependent with the heat capacity C_v_. For convenience the partition function is simplified by introducing
(23)$$Q(T)=e^{\frac{-\left[\Delta E_{0}+C_{v}(T-T_{m})\right]}{R}\left(\frac{1}{T}-\frac{1}{T_{m}}\right)}$$
leading to
(24)Z(T)=(1+Q(T))

The fraction of unfolded protein Θ_S_(T) is
(25)$$\Theta_{S}^{}(T)=\frac{\partial \ln Z(T)}{\partial \ln Q}=\frac{Q(T)}{1+Q(T)}$$

Θ_S_(T) is included for completeness only. It is not needed to calculate thermodynamic functions. The partition function Z(T) suffices to predict all thermodynamic properties.

The statistical mechanical two-state model is a continuous canonical partition function with only two states. It can be described as the non-cooperative limit of the multistate cooperative unfolding model (discussed below, Equation (29)) with a cooperativity parameter σ = 1 and only a single species of participating particles, N = 1. Figure 4 displays the predictions of Equations (18)–(22).

The native protein is the reference state with all thermodynamic functions being zero. In particular, the free energy change ΔF(T) is zero at 20 °C and becomes slightly negative, but never positive, between T_m_ and T_cold_. ΔF(T) decreases rapidly for temperatures T > T_m_ and T < T_cold_ according to
(26)$$\Delta F(T)≃(E_{0}+C_{v}(T-T_{m})\left(1-\frac{T}{T_{m}}\right)$$

The free energy ΔF(T) of the statistical mechanical two-state model again displays a trapezoidal temperature profile. Cold denaturation takes place at
(27)$$T_{\mathrm{cold}}=T_{m}-\frac{\Delta E_{0}^{}}{C_{v}^{}}$$

ΔE_0_ and C_v_ have opposite effects on T_cold_. Increasing $\Delta E_{0}^{}$ lowers T_cold_, increasing C_v_ leads to an upward shift.

Figure 4D compares the heat capacities predicted by the statistical-mechanical two-state model and the chemical equilibrium two-state model. The high-temperature peaks of the two models overlap precisely, but cold denaturation occurs at different temperatures. A discussion of other differences will follow in connection with the protein examples discussed below.

In summary, the statistical mechanical two-state model makes no assumption about the entropy. Additionally, as mentioned before, no weighting function Θ_U_(T) is needed to correctly calculate, the thermodynamic properties.

(c)Multistate Cooperative Unfolding Model [13]:

The energy of a system with Ν particles is characterized by its partition function
(28)$$Z(T)=\sum_{i}g_{i}e^{\frac{-\varepsilon_{i}}{k_{B}T}}$$

The partition function is the sum of exponential terms (Boltzmann factor) over all energy levels ε_i_, multiplied with their degeneracies g_i_, where k_B_ is the Boltzmann constant. The partition function determines the thermodynamic properties of the system (Equations (18)–(21) [28,29]. Here we use the partition function of the multistate cooperative Zimm–Bragg theory, originally developed for the α-helix-to-coil transition of polypeptides [1,30,31]. Its application to protein unfolding has been discussed recently [13]. The Zimm–Bragg theory has been applied successfully to the unfolding of helical and globular proteins of different structure and size [13,14,20,21,32,33,34,35,36]. Here, we use the equations given in [13].
(29)$$Z(T)=\left(\begin{array}{cc} 1 & 0 \end{array}\right){\left(\begin{array}{cc} 1 & \sigma q(T)\\ 1 & q(T) \end{array}\right)}^{N}\left(\begin{array}{c} 1\\ 1 \end{array}\right)$$
(30)$$q(T)=e^{\frac{-h(T)}{R}\left(\frac{1}{T}-\frac{1}{T_{0}}\right)}$$
(31)$$h(T)=h_{0}+c_{v}(T-T_{m})$$

h_0_ is the enthalpy change of the native → unfolded transition of a single amino acid residue. h_0_ is temperature-dependent with the heat capacity c_v_. Ν is the number of amino acids participating in the transition. The cooperativity parameter σ determines the sharpness of the transition. The σ parameter is typically in the range of 10^−3^–10^−7^. T_0_ is a fit parameter to shift the position of the heat capacity peak. T_0_ is usually close to T_m_. The temperature difference between heat and cold denaturation is $\Delta T\approx \frac{h_{0}}{c_{v}}$.

Figure 5 shows the thermodynamic temperature profiles predicted by Equation (29) in combination with Equations (18)–(21). Sigmoidal shapes are predicted for inner energy and entropy, and a trapezoidal shape for the free energy. Figure 5 is very similar to Figure 4 of the statistical-mechanical two-state model but is calculated with molecular parameters only. In particular, the free energy is again zero or negative, but never positive.

Figure 5 includes the DSC data of lysozyme heat denaturation (Figure 1). An excellent agreement between theory and experiment is obtained.

The multistate cooperative model describes protein unfolding with molecular parameters of well-defined physical meaning. In contrast, two-state models provide macroscopic parameters. Equation (29) can be applied to proteins of any size, including large antibodies with ~1200 amino acid residues and unfolding enthalpies of ~1000 kcal/mol [13].

## 3. DSC Experiments Compared to the Three New Protein Unfolding Models

### 3.1. Lysozyme Heat Unfolding

Figure 6 compares the experimental date of lysozyme unfolding with two unfolding models. The heat capacity C_p_(T) maximum is at T_m_ = 62 °C and the heat capacity increase is $\Delta C_{p}^{0}$ = 2.27 kcal/molK, in agreement with literature data of $\Delta C_{p}^{0}$ = 1.54–2.27 kcal/mol [10,18,20,22,23,37,38]. The red lines in Figure 6 represent the statistical-mechanical two-state model and were calculated with ΔE_0_ = 110 kcal/mol and C_v_ = 1.05 kcal/molK. A perfect fit of the experimental data is achieved.

The magenta dotted lines represent the weighted chemical equilibrium two-state model. The agreement with the experimental data is also very good. However, a small difference to the experimental data is observed near the midpoint of unfolding. Figure 7 displays an enlarged View of this region. DSC reports a zero free energy change for the native lysozyme. The free energy change becomes immediately negative upon unfolding. This result is correctly reproduced by the statistical-mechanical two-state model. In contrast, the Θ_U_(T)-weighted chemical equilibrium two-state model (Equation (17)) predicts little spikes of positive free energy at temperatures just before the midpoints of unfolding. The spikes are not observed experimentally. Although they are small and of no practical importance, they signify a thermodynamic discrepancy. The difference between experiment and model would be much larger if the parabolic free energy profile (Equation (10), Figure 2C) would be included. The multi-state cooperative model as described in Figure 5 also yields a perfect fit of the lysozyme DSC experiment [13].

In summary, three different models provide a good to excellent description of lysozyme DSC unfolding. At the midpoint temperature T_m_ all three models predict the extent of unfolding exactly as Θ_U_ = ½. Native and unfolded protein occur at equal concentrations. The free energy change is not zero, but DSC reports ΔG_DSC_(T_m_) = −0.756 kcal/mol. Indeed, a negative free energy is intuitively plausible as the protein is partially denatured at T_m_. This result is supported by two theoretical models. The statistical-mechanical two-state model yields ΔF(T_m_) = −0.462 kcal/mol, the multistate cooperative model yields ΔF(T_m_) = −0.855 kcal/mol. In contrast, the Θ_U_(T)-weighted chemical equilibrium two-state model predicts ΔG_Θ_(T_m_) = 0 kcal/mol.

### 3.2. β-Lactoglobulin Cold and Heat Denaturation

Bovine β-lactoglobulin (MW 18.4 kDa, 162 aa) folds up into an 8-stranded, antiparallel β-barrel with a 3-turn α-helix on the outer surface. β-Lactoglobulin in buffer without urea displays only heat denaturation (black squares in Figure 8A, data taken from Figure 1 of reference [39]). Unfolding takes place in the range of 55 °C < T < 96 °C with the C_p_(T) maximum at 78 °C. The unfolding enthalpy is ΔH_DSC_ = 74.5 kcal/mol (cf. ref. [39], Table 1, 0 M urea), the entropy ΔS_DSC_ = 0.213 kcal/molK, and ΔH_DSC_/ΔS_DSC_ = 349 K = 76 °C, consistent with the C_p_(T) maximum.

The Θ_U_(T)-weighted chemical equilibrium model, the statistical-mechanical two-state model, and the multistate cooperative model overlap completely as far as the heat capacity is concerned. On the other hand, some small differences are observed for enthalpy, entropy and free energy. The most conspicuous difference is seen for the free energy. The Θ_U_(T)-weighted chemical equilibrium model predicts a small peak of positive free energy for the native protein, which is not confirmed by the DSC experiment.

The conformational enthalpy is ΔH_0_ ≅ ΔE_0_ = 50–55 kcal/mol, which is small for a protein with 162 amino acid residues. Likewise, the molecular enthalpy parameter h_0_ = 380 cal/mol is also small compared to the typical 900–1300 cal/mol of globular proteins [20]. The molecular origin of the small unfolding enthalpy of β-lactoglobulin can be traced back to its extensive β-structure content. The enthalpy h_0β_ for the reaction β-structure → disordered amino acid was measured as h_0β_ = 230 cal/mol in a membrane environment [40].

A protein can also be unfolded by cooling. Cold denaturation usually occurs at subzero temperatures but can be shifted to above 0 °C by high concentrations of chemical denaturant or extreme pH values. All models discussed above, predict cold denaturation, provided the heat capacities $\Delta C_{p}^{0}$ or C_v_ are non-zero. In fact, the temperature difference between heat and cold denaturation depends strictly on the ratio of conformational enthalpy/heat capacity.

Only a few DSC experiments showing at least partial cold denaturation are available. One of the best examples is DSC-unfolding of β-lactoglobulin in 2.0 M urea solution [39].

The DSC data are taken from Figure 2 of reference [39]. The simulation of C_p_(T) is shown in Figure 9A for the Θ_U_(T)-weighted chemical equilibrium model and in Figure 9B for the two statistical models. The DSC experiment begins at −9 °C where the protein is in a disordered state. Heating induces a disorder → order transition with a heat capacity maximum at 4 °C, a folding enthalpy ΔH_DSC_ = 78.3 kcal/mol and an entropy change of ΔS_DSC_ = 0.282 kcal/molK. The ratio ΔH_DSC_/ΔS_DSC_ is 277K = 4 °C, consistent with the heat capacity maximum.

At ~18–30 °C the protein is in the native-like conformation. Cooling reverses the before mentioned process and the protein returns to a disordered conformation with a simultaneous release −78 kcal/mol (cf. Figure 9C). The heat capacity peak at 4 °C is hence the mirror image of cold denaturation (see Figure 2 in [39]. Heating β-lactoglobulin above 30 °C destroys the native structure. The order → disorder transition has a heat capacity maximum at 57 °C, ΔH_DSC_ = 104 kcal/mol and ΔS_DSC_ = 0.312 kcal/molK. The ratio ΔH_DSC_/ΔS_DSC_ of heat denaturation is 333 K = 60 °C.

β-Lactoglobulin is less stable in urea solution as the midpoint temperature T_m_ is shifted from 78 °C to 57 °C. Such a decrease in temperature is common in chemical denaturants. However, it is usually associated with a decrease in enthalpy, not an increase [35,36,41]. In the present case the enthalpy increases by ~40%, the entropy by ~50%, resulting, in turn, in a 20 °C downshift of T_m_.

The DSC data of β-lactoglobulin were analyzed with the Θ_U_(T)-weighted chemical equilibrium model and the two statistical models. The conformational enthalpy ΔH_0_ is equal to the inner energy ΔE_0_ with ΔH_0_
≈ ΔE_0_ = 56 kcal/mol. These parameters are also identical to those obtained for β-lactoglobulin in buffer. In contrast, the heat capacities $\Delta C_{p}^{0}$, C_v_ and c_v_ are 2–3 times as large, most likely due to the binding of urea molecules.

The three models discussed above provide a good simulation of all the experimental data. In particular, they reproduce the trapezoidal temperature profile of the free energy (black squares in Figure 9D). The native protein has a zero heat capacity change and, in turn, a zero free energy change. Unfolding leads to negative free energies, both for heat and cold denaturation. The two statistical models perfectly fit these experimental data. The Θ_U_(T)-weighted chemical equilibrium two-state model displays small positive peaks in the vicinity of T_m_ and T_cold_, which are not supported by DSC.

## 4. Conclusions

Under equilibrium conditions protein stability is determined by (i) the midpoint of heat denaturation, (ii) the temperature difference between heat and cold denaturation, and (iii) the width and cooperativity of the unfolding transitions. These parameters are intimately connected to the thermodynamic properties of the system. The thermodynamics of protein unfolding is completely characterized by the temperature profiles of enthalpy, entropy and free energy. The building stone of these thermodynamic properties is the heat capacity C_p_(T), which can be measured precisely by differential scanning calorimetry. In this review we have emphasized the almost completely ignored concept of the experimental and model-independent evaluation of the heat capacity in terms of the thermodynamic functions ΔH(T), ΔS(T) and ΔG(T). The evaluation is straightforward and simple. It is hence quite surprising that this approach has not been considered in the relevant literature.

Thermodynamic unfolding models should predict not only the heat capacity C_p_(T), but also the complete set of thermodynamic functions. The DSC experiment reveals a sigmoidal temperature profiles for enthalpy and entropy and a trapezoidal profile for the free energy. Focusing on the unfolding transition proper, the heat capacity change of the native protein is zero and all the changes of the thermodynamic functions are equally zero. No positive free energy is measured for the native protein, which is in contrast to the prediction of the prevailing chemical equilibrium two-state model. Experimental temperature profiles were shown for the heat-induced unfolding of the globular protein lysozyme and for the heat and cold denaturation of the β-barrel protein β-lactoglobulin. The experimental results are compared to the predictions of four different models, that is, two chemical equilibrium two-state models and two statistical mechanical models. All four models described the heat capacity equally well. The currently prevailing chemical equilibrium two-state model fails however in simulating the thermodynamic temperature profiles. This model was therefore modified by multiplying all thermodynamic functions with the extent of unfolding, yielding the Θ_U_(T)-weighted chemical equilibrium model. This is an empirical approach which fits all of the thermodynamic data quite well but displays a small discrepancy with respect to the experimental results in the vicinity of the unfolding transition. The two statistical-mechanical models have a rigorous thermodynamic foundation and avoid this difficulty. They provide the best simulations of the experimental data.

Two-state models are non-cooperative approximations to a cooperative multistate protein folding ⇄ unfolding equilibrium. They describe protein unfolding in terms of two macroscopic parameters, the conformational enthalpy ΔH_0_
≃ ΔE_0_ and the heat capacity $\Delta C_{p}^{0}$~2 C_v_. In contrast, the multistate cooperative unfolding model uses molecular parameters, that is, the enthalpy h_0_ per amino acid residue, the heat capacity c_v_, the cooperativity parameter σ, and N, the number of participating amino acid residues. The multistate cooperative model can be applied to proteins of any length, e.g., antibodies with 1200 amino acid residues and unfolding enthalpies of 1000 kcal/mol [13,36]. In contrast, two-state unfolding models are best suited for unfolding enthalpies of 50–200 kcal/mol, typically found for small proteins only.

Protein unfolding is characterized by large enthalpies and entropies, but small free energies (enthalpy-entropy compensation), as discussed in detail by comparing lysozyme and β-lactoglobulin. The free energy is not a good criterion to judge protein stability. Better parameters are defined above. The Θ_U_(T)-weighted chemical equilibrium two-state model, the statistical-mechanical two-state model, and the multistate cooperative model provide quantitative thermodynamic interpretations of these parameters.

## Figures and Tables

**Figure 1 ijms-24-05457-f001:**
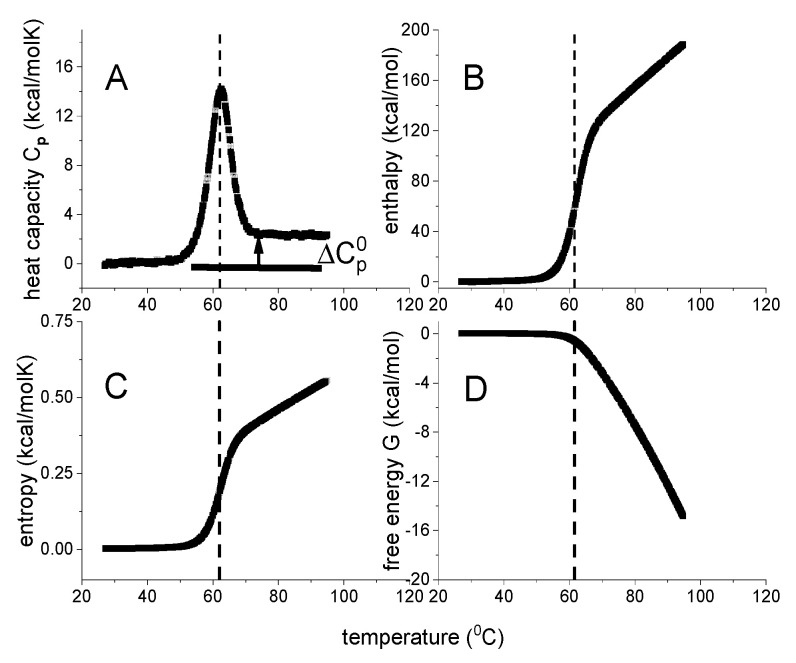
DSC of lysozyme. Model–independent thermodynamic analysis (50 µM, 20% glycine buffer, pH 2.5). (**A**) Heat capacity. DSC data (temperature resolution 0.17 °C) taken from reference [19,20]. (**B**) Enthalpy ΔH_DSC_(T) (Equation (1)). (**C**) Entropy ΔS_DSC_(T) (Equation (2)). (**D**) Gibbs free energy ΔG_DSC_(T) (Equation (3)).

**Figure 2 ijms-24-05457-f002:**
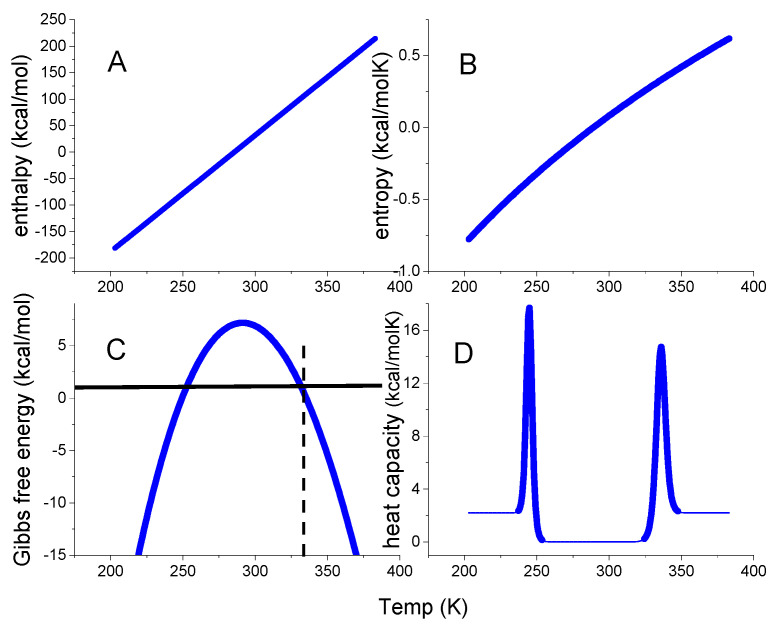
The thermodynamic functions of the standard chemical equilibrium two–state model calculated with the parameters typical for lysozyme, ΔH_0_ = 110 kcal/mol and $\Delta C_{p}^{0}$ = 2.2 kcal/molK. Dashed vertical line: midpoint temperature T_m_ = 62 °C. (**A**) Enthalpy ΔH_NU_(T) (Equation (8)). (**B**) Entropy ΔS_NU_(T) (Equation (9)). (**C**) Gibbs free energy ΔG_NU_(T) (Equation (10)). (**D**) Heat capacity C_p_(T) (Equation (12)).

**Figure 3 ijms-24-05457-f003:**
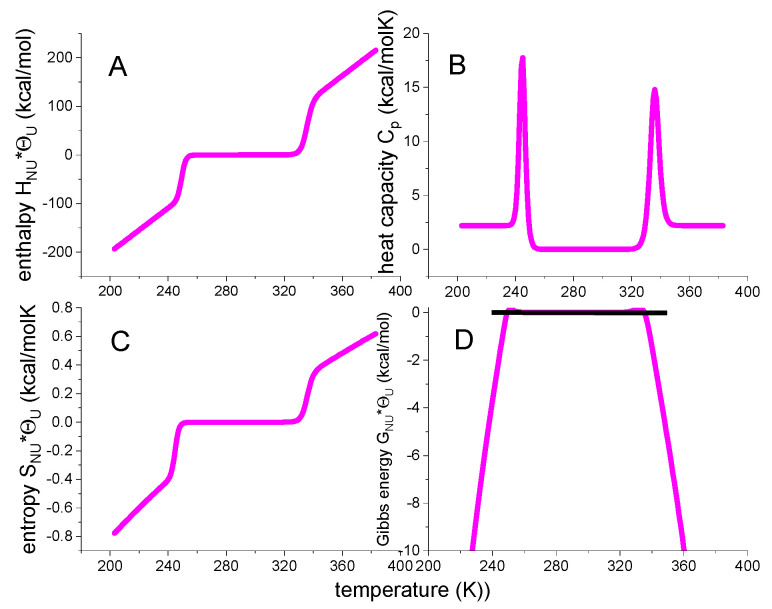
Θ_U_(T)-weighted chemical equilibrium two-state model. (**A**) Enthalpy (Equation (15)). (**B**) Heat capacity (Equation (12)). (**C**) Entropy (Equation (16)). (**D**) Free energy (Equation (17)). Fit parameters: ΔH_0_ = 107 kcal/mol, $\Delta C_{p}^{0}$ = 2.27 kcal/molK.

**Figure 4 ijms-24-05457-f004:**
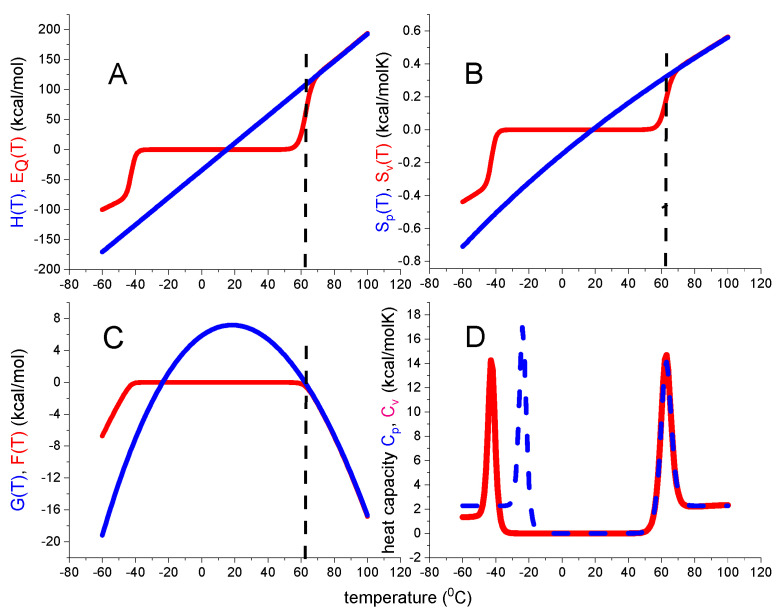
Statistical–mechanical two-state model. Red lines calculated with ΔE_0_ = 110 kcal/mol and C_v_ = 1.05 kcal/molK. (**A**) Inner energy ΔE(T). (**B**) Entropy ΔS_v_(T). (**C**) Helmholtz energy ΔF(T). (**D**) Heat capacity C_p_(T). Blue lines are calculated with the standard chemical equilibrium two-state model, using, ΔH_0_ = 107 kcal/mol, $\Delta C_{p}^{0}$ = 2.27 kcal/molK. Dashed vertical lines: midpoint temperature T_m_ = 62 °C.

**Figure 5 ijms-24-05457-f005:**
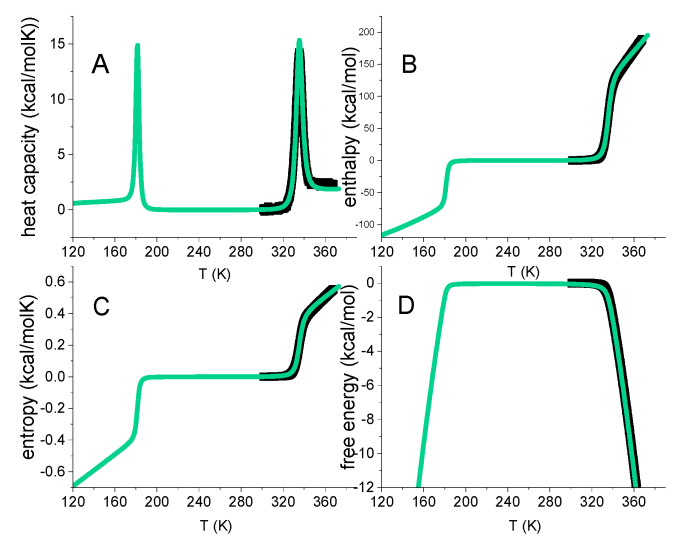
Multistate cooperative model. Green lines calculated with: N = 129 amino acid residues. h_0_ = 900 cal/mol. c_v_ = 7 cal/molK. Cooperativity parameter σ = 5 × 10^−7^. Black data points: thermal unfolding of lysozyme measured with DSC. Same data as in Figure 1.

**Figure 6 ijms-24-05457-f006:**
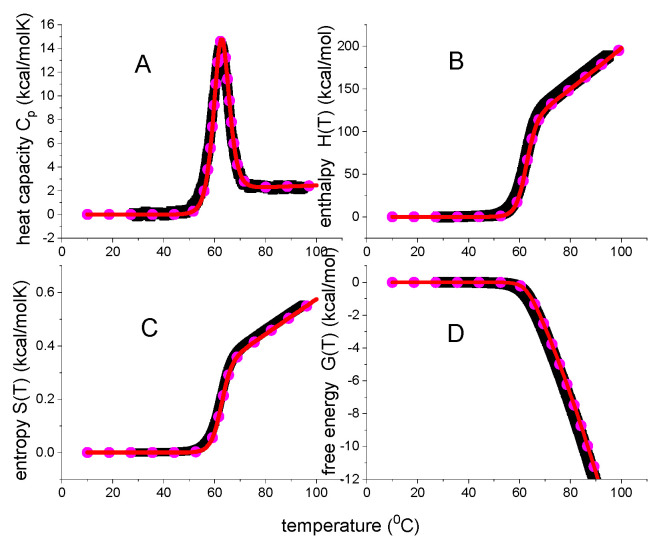
Analysis of lysozyme heat unfolding with 2-state models. Black data points: DSC data of Figure 1. Red lines: statistical-mechanical two-state model (ΔE_0_ = 110 kcal/mol and C_v_ = 1.05 kcal/molK). Magenta dotted line: Θ_U_(T)-weighted chemical two-state model (ΔH_0_ = 107 kcal/mol, $\Delta C_{p}^{0}$ = 2.27 kcal/molK).(**A**) Heat capacity. (**B**) Enthalpy. (**C**) Entropy. (**D**) Free energy.

**Figure 7 ijms-24-05457-f007:**
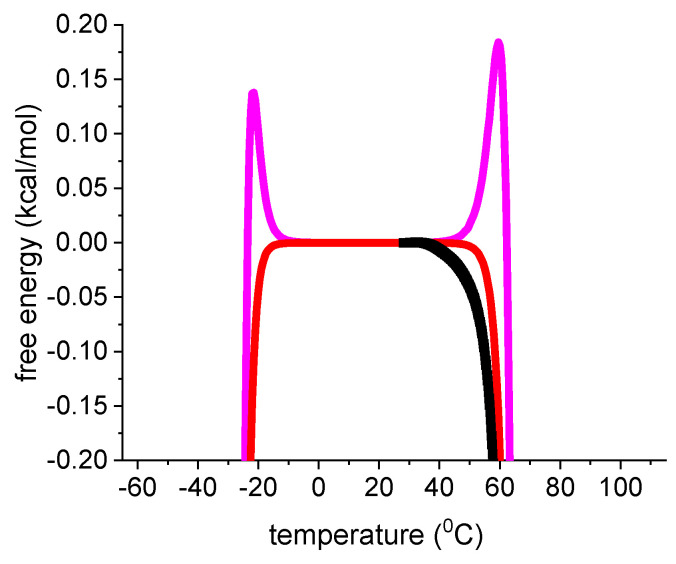
Enlarged view of the free energy. Black data points: DSC results for lysozyme heat unfolding. Red line: statistical mechanical two-state model (ΔE_0_ = 110 kcal/mol, C_v_ = 1.05 kcal/mol). Magenta line: Θ_U_(T)-weighted chemical equilibrium two-state model (Equation (17), ΔH_0_ = 1107 kcal/mol, $\Delta C_{p}^{0}$ = 2.27 kcal/molK).

**Figure 8 ijms-24-05457-f008:**
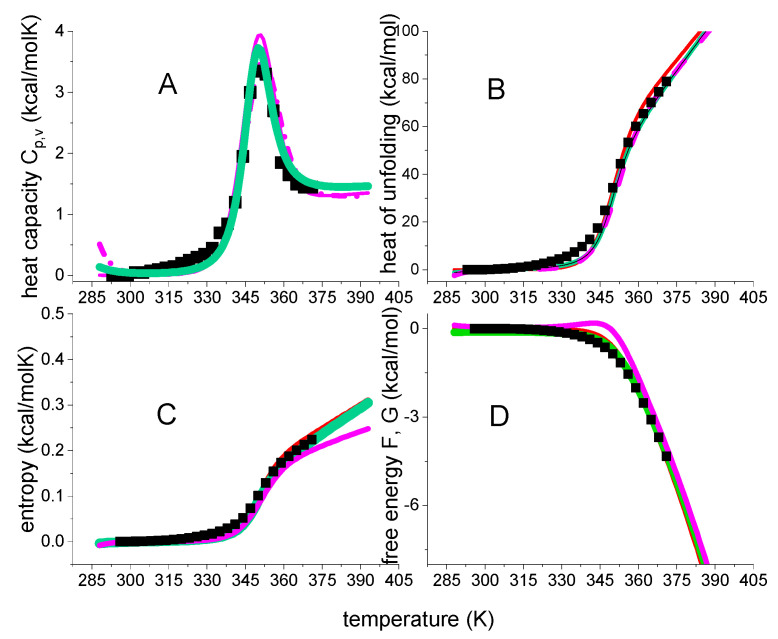
DSC of β-lactoglobulin in 0.1 M KCl/HCl, pH 2.0 buffer. Black data points in panel A are taken from reference [39] (Figure 1). Magenta lines: Θ_U_(T)-weighted chemical equilibrium two-state model (ΔH_0_ = 50 kcal/mol, $\Delta C_{p}^{0}$ = 1.3 kcal/molK). Red lines: statistical-mechanical two-state model (ΔE_0_ = 55 kcal/mol, C_v_ = 0.6 kcal/molK) Green lines: multistate cooperative model (h_0_ = 380 cal/mol, c_v_ = 3 cal/molK, σ = 7 × 10^−5^, N = 160). (**A**) Heat capacity. (**B**) Unfolding enthalpy. (**C**) Unfolding entropy). (**D**) Free energy of unfolding.

**Figure 9 ijms-24-05457-f009:**
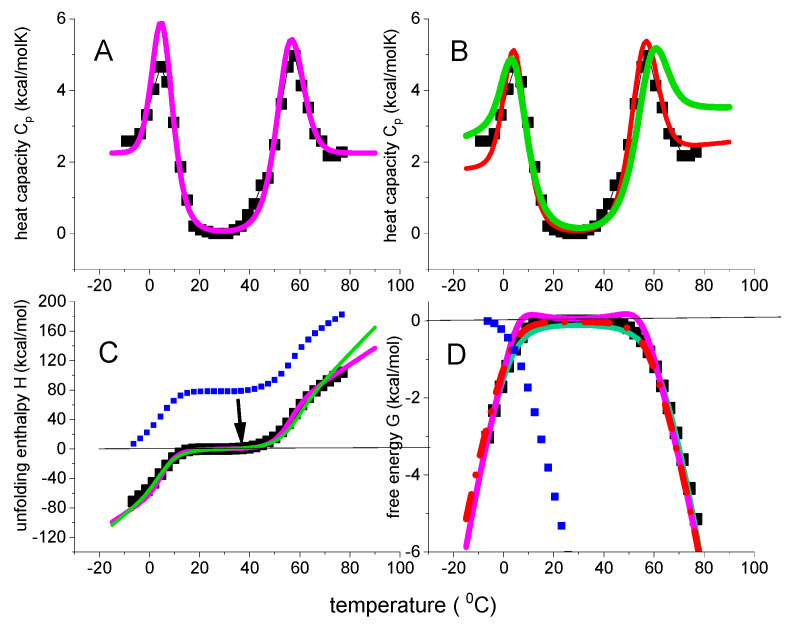
Thermal folding and unfolding of β-lactoglobulin in 2.0 M urea solution. Magenta lines: Θ_U_(T)-weighted chemical equilibrium model. ΔH_0_ = 56 kcal/mol; $\Delta C_{p}^{0}$ = 2.4 kcal/moKl. Red lines: statistical-mechanical two-state model. ΔΕ_0_ = 55 kcal/mol; C_v_ = 1.2 kcal/molK; Green line: multistate cooperative model. h_0_ = 0.58 kcal/mol, c_v_ = 13 cal/molK, σ = 6 × 10^−5^, N = 80. (**A**) DSC heat capacity data taken from reference [39]. Simulation with the Θ_U_(T)-weighted chemical equilibrium model. (**B**) Same DSC data as in panel A. Simulations with the statistical models (**C**) Enthalpy ΔH_DSC_(T). Integration of the C_p_(T) data according to equation 1 generates the blue data points. The data are then shifted by −78.3 kcal/mol, the enthalpy released upon cold denaturation, resulting in the black data points. This scale-shift assigns a zero enthalpy to the native protein. (**D**) Free energy. Evaluation of C_p_(T) according to equations 1–3 leads to the blue data points. A related scale-shift for the entropy as for the enthalpy (not shown) and recalculation of the free energy results in the black data points. The free energy change of the native protein is now zero. Cold and heat denaturation generate negative free energies (for more details see reference [15]).

## Data Availability

Published data (references given).

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
