# Peer review of "Protein Unfolding—Thermodynamic Perspectives and Unfolding Models"

_ijms, 2023, doi:10.3390/ijms24065457_

Round 1

Reviewer 1 Report

This is a fine, clearly written paper that cerianly deserves to be published. The authors start by stating that the standard two-state model of protein denaturation assumes a chemical equilibrium between only two protein conformations, the native protein (N) and the unfolded protein (U) and fits the heat capacity Cp(T) quite well, but fails in simulating the other thermodynamic properties. In this connection the authors stress the direct and model-independent evaluation of the heat capacity in terms of the thermodynamic functions H(T), S(T) and G(T). The authors then present a new statistical-mechanical two-state model based on a two-parameter partition function from which all thermodynamic parameters can be derived. The new model predicts a zero free energy for the native protein, which is confirmed experimentally by DSC. They state that according to basic thermodynamics it follows that all thermodynamic properties must also be zero for Cp = 0 cal/molK if the heat capacity is zero. It seems to me that is only the case for absolute zero, not for general temperature and I would like the authors to comment on this. Indeed this seems to be a variant of the Nernst theorem with a zero heat capacity that leads to zero values for all thermodynamic functions. In addition Eq. 22 contains only macroscopic parameters, no microscopic parameters and yet it is claimed to be a statistical mechanical theory. Please comment on this. The authors refer to "simulations" but I think they actually mean "calculations". Simulation is something different. The authors should answer the above querries in a revised version of the MS>

Reviewer 2 Report

The goal of the manuscript is to emphasize the analysis of enthalpy, entropy, and free energy, which all can be derived from the measured heat capacity in the DSC experiment. The authors should communicate the rationale for such an analysis better. It seems like the analysis of the derived parameters may be helpful in the selection of a model, describing the results of the DSC experiment, by examining derived parameters for non-physical features. For example, the weighted model gives positive spikes of free energy in both considered examples, and the authors say that this is “not supported by DSC”. Such a claim should be explained in detail since this is the main theme of the manuscript.

The manuscript suffers from multiple typos and randomized punctuation. Some English editing will improve the manuscript (check the use of the word “proper”). There are redundancies in the abstract, equations 1 and 2 should be checked on the parentheses use, and equation 3 likely misses deltas. It is surprising to see the parameter of the extent of unfolding in an equation in section 3.1.2 describing the standard model, when the use of this parameter is emphasized in section 3.1.3, please explain. In section 3.2 please explain the difference between Cp and Cv, and the physical meaning of the inner energy. It would be nice to define R in equations 18-23.

In conclusion, the manuscript is worth to be published, it just needs some attention to detail.
